# Most Short Children with Cystic Fibrosis Do Not Catch Up by Adulthood

**DOI:** 10.3390/nu13124414

**Published:** 2021-12-10

**Authors:** Margaret P. Marks, Sonya L. Heltshe, Arthur Baines, Bonnie W. Ramsey, Lucas R. Hoffman, Michael S. Stalvey

**Affiliations:** 1Department of Pediatrics, University of Alabama, Birmingham, AL 35233, USA; mpmarks@uabmc.edu; 2Cystic Fibrosis Research Center, University of Alabama, Birmingham, AL 35233, USA; 3Department of Pediatrics, University of Washington, Seattle, WA 98105, USA; sonya.heltshe@seattlechildrens.org (S.L.H.); bonnie.ramsey@seattlechildrens.org (B.W.R.); lhoffm@uw.edu (L.R.H.); 4CFF TDNCC, Seattle Children’s Research Institute, Seattle, WA 98121, USA; arthur.baines@seattlechildrens.org; 5Department of Microbiology, University of Washington, Seattle, WA 98105, USA

**Keywords:** cystic fibrosis, growth restriction, final height

## Abstract

Poor linear growth is common in children with cystic fibrosis (CF) and predicts pulmonary status and mortality. Growth impairment develops in infancy, prior to pulmonary decline and despite aggressive nutritional measures. We hypothesized that growth restriction during early childhood in CF is associated with reduced adult height. We used the Cystic Fibrosis Foundation (CFF) patient registry to identify CF adults between 2011 and 2015 (ages 18–19 y, *n* = 3655) and had height for age (HFA) records between ages 2 and 4 y. We found that only 26% CF adults were ≥median HFA and 25% were <10th percentile. Between 2 and 4 years, those with height < 10th percentile had increased odds of being <10th percentile in adulthood compared to children ≥ 10th percentile (OR = 7.7). Of HFA measured between the 10th and 25th percentiles at ages 2–4, 58% were <25th percentile as adults. Only 13% between the 10th and 25th percentile HFA at age 2–4 years were >50th percentile as adults. Maximum height between ages 2 and 4 highly correlated with adult height. These results demonstrate that low early childhood CF height correlates with height in adulthood. Since linear growth correlates with lung growth, identifying both risk factors and interventions for growth failure (nutritional support, confounders of clinical care, and potential endocrine involvement) could lead to improved overall health.

## 1. Introduction

Cystic fibrosis (CF) is an autosomal recessive disease caused by mutations in the cystic fibrosis transmembrane conductance regulator (CFTR) gene. CFTR encodes a chloride channel expressed in a variety of epithelial tissues, and mutations in this gene impair protein activity and/or expression [1]. CFTR protein dysfunction affects multiple organ systems and unequivocally contributes to growth deficits and progressive lung disease from birth and throughout infancy and childhood [2]. The association between nutrition and pulmonary outcomes in CF is well established, particularly when using weight-for-length (WFL) and body mass index (BMI) as key nutritional indices [3,4].

The CF Foundation recommends that children ages 0 to 2 years maintain their WFL at or above the 50th percentile, and that children ages 2 to 19 years maintain their BMI at or above the 50th percentile [4]. The advent of universal newborn screening for CF has allowed for earlier diagnosis and earlier interventions to improve nutritional status. For example, the Wisconsin CF Neonatal Screening Project was a randomized controlled trial initiated in 1984 that examined longitudinal nutritional outcomes in patients diagnosed with CF via newborn screening compared to control patients diagnosed with CF later in infancy. Weight-for-age (WFA) and height-for-age (HFA) Z-scores were significantly better throughout childhood and adolescence in patients diagnosed by newborn screening compared to controls. Although growth metrics in control CF participants improved following diagnosis, they never reached levels comparable to those diagnosed by newborn screening. This discrepancy was particularly notable in HFA Z-scores, indicating that early-life growth stunting associated with delayed CF diagnosis may have lasting effects on stature [5].

Strong correlations between WFA percentiles in early life and subsequent pulmonary status have been reported in cross-sectional studies [6,7]. Recently, a longitudinal study by Sanders et al. examining early life growth trajectories showed that infants and children with CF who maintained their WFL and BMI consistently above the 50th percentile had the best percent-predicted forced expiratory volume in 1 s (ppFEV1) at ages 6–7. Furthermore, infants who entered the study with a WFL < 50th percentile, but who eventually achieved a WFL ≥ 50th percentile before age 2, had higher ppFEV1 at age 6 years compared to children who did not achieve BMI ≥ 50th percentile until after age 2 [3]. Similar to WFL and BMI, linear growth is increasingly recognized as an important prognosticator of respiratory morbidity and mortality in CF. A longitudinal study by Assael et al. reported that people with CF who developed severe respiratory compromise had reduced early-life growth velocity, even before any appreciable decline in lung function. Thus, impaired linear growth is an early indicator of pulmonary disease severity that manifests before spirometry can be reliably performed [8]. The importance of early-life growth trajectories was also demonstrated in another study by Sanders et al., who reported that children who maintain HFA above the 50th percentile between diagnosis of CF and age 6–7 years have the highest pulmonary function at age 6–7 years, irrespective of their BMI percentile [9]. Furthermore, stunting, defined as a height below the 5th percentile, is an independent predictor of mortality [10].

These data suggest that improving nutrition early in life may also improve CF lung disease outcomes, and perhaps overall survival, underscoring the importance of stature in the prognosis and perhaps pathogenesis of CF lung disease. Based on data from the US CF Foundation Patient Registry (CFFPR), meticulous attention to nutritional interventions in both early life and throughout childhood and adolescence is successful in achieving the goals of WFL and BMI targets of above the 50th percentile. Over the past 15 years, the median BMI percentile in individuals 2–19 years has improved from 45.1% to 58.3%. Further, the number of children with WFA <10th percentile has improved from 20.9% to 10.1%. Additionally, use of supplemental feeding tubes has increased from 8.8% to 10.3%. However, HFA less than the fifth percentile has only improved from 14.6% to 9.5%. Presently, the median BMI for adults with CF over the age of 20 is 23.0. For CF individuals between 2 and 19 years, the median weight percentile is 49.3%, but the height percentile is only 38.4%. In children less than 24 months old, the median WFL percentile is 63.7%, the weight percentile is 42.8%, and—consistent with the above—the length percentile is 30.2% [11]. Thus, risk factors for short stature itself remain to be defined. We hypothesized that impaired early linear growth would correlate with stunting as an adult among people with CF.

To test this hypothesis, we used CFFPR data to determine whether linear growth impairment in early life in CF is associated with a reduced final adult height. Given the importance of short stature as a prognosticator of morbidity and mortality in CF, the early identification of children at risk for stunting and appropriate interventions (nutritional support, potentially mediating confounders from clinical care, and endocrine involvement) could improve overall health outcomes in patients with CF.

## 2. Materials and Methods

The United States CFFPR was used for this retrospective case–control study. People with CF who were age 18–19 years between 2011 and 2015 with any height measurements were selected. Then, the height measurements for those young adults were retrospectively ascertained from the CFFPR from when they were between 2 and 4 years of age. CFFPR provides one annualized height measurement per person per year calculated by averaging the maximum value per quarter. Heights were standardized to US Centers for Disease Control (CDC) height for age percentiles (HFA%) [12]. For a given individual, the lowest height percentile of the two recorded at 18 and 19 years of age in the CFFPR defined their categorization into groups: <10th, 10th–25th, 25th–50th, or ≥50th HFA%. Similarly, the lowest annualized HFA%, up to three recorded for each person between 2 and 4 years of age, defined their inclusion into early childhood HFA groups. As a sensitivity analysis, children who’s annualized HFA% < 10 in all three years between 2 and 4 years of age were examined. Summary statistics, odds ratios (OR), and Pearson correlation coefficients with 95% confidence intervals (CI) are reported. The Seattle Children’s Hospital (Seattle, WA, USA) Institutional Review Board granted approval for this research. SAS version 9.2 (Cary, NC, USA) or R version 3.3.3 (R Foundation for Statistical Computing, Vienna, Austria) were used for all analyses.

## 3. Results

### Early Childhood Height for Age Is Associated with Adult Height

Among the 3655 individuals with CF aged 18–19 between 2011 and 2015 in the CFFPR, only 26% (*n* = 939) were at or above the median height for their age (50th HFA%) and 25% (*n* = 915) had an HFA% < 10 in adulthood (Table 1). Between the ages of 2 and 4 years, 28% (1034/3566) of those same children had at least one annualized HFA% < 10, and 54% of those short children went on to be <10 HFA% in early adulthood (increased odds of 7.7 (95% CI = 6.5–9.1) compared to children with CF between 2 and 4 years with HFA% ≥ 10). Among the children with HFA% < 10 in all three years between 2 and 4 years, 67% were short as adults (<10 HFA%) and had much higher odds of being so compared to children between 2 and 4 years of age who were not always <10 HFA% in early childhood (OR = 9.0 (95% CI = 7.3–11.0)). Only 13% of children with HFA% between 10 and 25 at between 2 and 4 years achieved or exceeded 50 HFA% as adults. The maximum height measurements between 2 and 4 years were highly correlated with maximum height at age 18–19 (r = 0.64, 95% CI = (0.62, 0.66)) (Figure 1).

## 4. Discussion

We found that early-life growth impairments of preschool children with CF correlated with stunting in early adulthood. Growth impairment in people with CF remains common and challenging to identify and address, although improvements in nutritional metrics have been observed in all ages in CF over the past several decades. According to the CF Foundation’s 2019 Patient Registry Annual Data Report, the median WFL among children with CF in the US ages 0 to 2 years was above the recommended 50th percentile, and the median BMI for children ages 2 to 19 years was above the recommended 50th percentile. However, relying solely on WFL and BMI as markers of nutritional status fails to identify children with suboptimal nutrition based on WFA and HFA percentiles, as children with stunting may maintain relatively normal WFL and BMI due to proportionally poor linear and weight growth. For example, despite the improvements in median WFL and BMI, both LFA and HFA percentiles in children with CF remain well below those of the general population [11]. A 2017 study by Konstan et al. examining growth parameters of 11,669 children with CF ages 2 to 18 years reported that 20.5% of participants whose BMI were at or above the recommended 50th percentile had HFA below the 10th percentile [13]. Growth trajectories of patients with CF diagnosed by newborn screening suggest that the stunting frequently observed in CF begins in early infancy and persists despite weight normalization. The Baby Observational and Nutrition Study (BONUS) examined the growth patterns of 231 infants diagnosed with CF via newborn screening compared to healthy cohorts. BONUS infants achieved normal weight by 12 months of age; however, their length lagged behind those of healthy peers [14]. As an aside, the consideration of BMI as the sole growth measurement would not reflect such stunting, and we therefore considered height and weight separately in this study.

In addition to early stunting, a “second hit” to the growth of children with CF may occur in puberty. Historically, studies have suggested that children with CF also have reduced height velocities (HV) throughout childhood that are particularly compromised during puberty. These studies used Tanner–Davies growth curves developed in the USA in 1985 to compare HV in children with CF to children without CF [15,16]. The Tanner–Davies curves were generated using longitudinal growth data from a restricted population of European descent with superimposed cross-sectional data from the US National Center for Health Statistics to establish ages of peak HV for children who matured early, average, and late [17]. More recently, reference data based on longitudinal growth metrics in a more diverse population of healthy youths in the USA allowed for the calculation of HV percentiles and Z-scores, which facilitates comparison between children with CF and healthy peers [18]. Using these contemporary data, Zysman-Colman et al. demonstrated that the HV percentiles of children with CF ages 5–17 years fall within the 25th to 75th percentiles of healthy children. Furthermore, shorter children with CF tended to have lower HV Z-scores than taller children with CF. This difference was most notable in pre-pubertal children, which suggests that final height is determined early in life in CF. The study cohort had below average lengths from birth that persisted into adulthood despite normal childhood and pubertal growth velocities and adequate nutrition [19].

The mechanism underlying growth restriction in CF remains unclear. Impaired CFTR activity has detrimental effects on multiple organ systems that could contribute to poor growth. Malnutrition can undoubtedly lead to poor growth; however, as BONUS demonstrated, stunting in CF begins in early infancy despite nutritional interventions that generally normalized weight achievement [14]. Infants included in the BONUS study with pancreatic insufficiency initiated pancreatic enzyme replacement therapy at a mean age of 2 months, with doses consistent with the CF Foundation guidelines. In the infants that were exclusively formula-fed, 40.2% received high calorie formula—greater than or equal to 24 kcal/oz—at 3 months, 52.4% at 6 months, and 49.0% at 12 months. Infants that were exclusively breastfed could not be assessed for total caloric intake; however, they weighed more than formula-fed or a combination of the two, at 3 months. This finding did not persist at 6 or 12 months of age. Interestingly, the feeding type (whether breast, formula-fed, or a combination of the two) was not associated with infant length during the BONUS study. The summation findings of BONUS demonstrated normalization of weight in CF infants by 12 months; however, despite these nutritional improvements, length did not normalize. Consistent with the studies previously mentioned, only 13.6% were less than the 10th percentile WFA, but 23.9% were less than the 10th percentile LFA.

Historically, providers (both pulmonary and endocrine) have attributed poor nutritional absorption and chronic disease as the primary influence on linear growth in CF. Similarly, lung disease could diminish nutritional outcomes. However, CF animal models have demonstrated growth restriction in the absence of these confounding variables. CF mice do not develop lung disease—and lung disease only manifests after 6 months of age in CF rats—although both exhibit impaired growth [20,21,22]. Data from these animal models combined with clinical data offer clues regarding other potential contributors to poor growth in CF. For example, both animals and people with CF have reduced anabolic drive secondary to lower tissue concentrations of insulin-like growth factor-1 (IGF-1) and insulin deficiency associated with CF-related diabetes [23,24,25,26].

One of the most remarkable findings suggestive of a primary defect in the growth axis unrelated to nutritional intake is the decreased IGF-1 concentrations seen in many animal models of CF, including newborn CF piglets (compared to non-CF littermates) [23]. CFTR-deficient mice and rats have reduced total body length and femur length as well as reduced serum IGF-1 concentrations. These findings suggest that CFTR dysfunction intrinsically affects the endocrine growth axis. Furthermore, growth-restricted CF rats exhibit growth plate alterations compared to controls, demonstrated by reduced hypertrophic chondrocyte volume as well as an overall reduction in growth plate thickness [22]. Similarly low IGF-1 was found in newborn blood spots of CF infants, which also suggests intrinsic defects in the growth axis and argues against intestinal malabsorption as the primary etiology of poor growth [23]. Further addressing the implication that nutritional intake is or is not the causative agent for decreased IGF-1 concentrations, a study by Hardin et al. investigated enteral nutritional supplementation versus enteral nutritional supplementation combined with treatment with human growth hormone (hGH). The authors evaluated outcomes in both growth and contribution to serum IGF-1 levels. Despite one year of enteral nutrition supplementation, subjects not on hGH therapy exhibited no change in serum IGF-1 concentrations. However, after the addition of hGH to the enteral feeding group—at year 2—there was an improvement [27].

Additional evidence of dysfunctional CFTR protein’s intrinsic effect on growth, as well as a potential source for improvement, is demonstrated by improved growth with CFTR modulator therapy, which increases the activity of the defective chloride channel in vivo. Stalvey et al. demonstrated this in a post hoc analysis of linear growth observed longitudinally in pre-pubertal children with CF and at least one copy of the G551D CFTR mutation who were enrolled in the G551D Observational Study (GOAL) of children initiating the CFTR modulator ivacaftor. At baseline, participants were below average in HFA. Six months following the initiation of ivacaftor, both HFA and HV significantly increased. In the placebo-controlled, randomized Evaluation of Efficacy and Safety of VX-770 in Children Six to Eleven Years Old with CF (ENVISION) study, there was continued improvement in HFA beyond 6 months in children treated with ivacaftor compared to controls [28]. Newer, highly effective modulators are now available for people with the more common F508del CFTR mutation, and ongoing studies are collecting growth data on pubertal and pre-pubertal children with CF following the initiation of treatment. As part of the outcomes studied in children and adults, these studies are evaluating changes in body composition. Other studies are underway to assess essential growth changes in early infancy and childhood (when growth rates tend to be relatively fast), as well as growth factors (such as IGF-1), and to monitor the influence of CFTR therapies as they are initiated. Investigators hope to provide insights into the biological–pathologic process of growth restriction, which begins in infancy and carries through to adulthood.

There are certain limitations in utilizing the CFFPR for our retrospective case–control analysis. The first would be how our population compares to healthy children growing at the 10th or below percentile at age 2–4 years. The authors do not argue that children—with or without CF—growing along that percentile may be growing as intended for their genetic potential. We are in fact attempting to illustrate and discredit the common misconception by providers, that poor growth in CF children early in life has time to “catch up” by adulthood. In our comparisons, we did not have heights from the parents to calculate mid-parental height targets for the subjects, but given the data utilized is obtained from the national registry, one would anticipate a near mean average adult height of the parents. Additionally, the recent report by Zysman-Colman et al. demonstrates that when parental heights are known, there is a reduction in obtainment of mid-parental linear growth targets in children with CF [19]. Our analysis would further suggest that CF children with evidence of poor growth early in life warrant closer attention. It has also been observed that the age of the onset of puberty was historically often delayed in children with CF, raising the possibility that some in our study population were still growing at the end of the study period [29,30,31]. While we did not analyze the effects of some potential confounders, including gender/sex, CFTR genotype, or differences in therapy (including CFTR modulator use, which was likely rare in this study population), future studies could address the effects of these covariates. Similarly, registry data could be used to construct predictive models based on early childhood growth characteristics of later growth outcomes, an analysis we did not perform here.

In summary, our study demonstrates that height during preschool ages correlates with subsequent adult height achievement in people with CF. Recognition of impairment in early life may be essential to improving outcomes. A team-based multi-disciplinary approach to these high-risk children by their pulmonologists, registered dietitians, social services, and consultative services (endocrine and gastroenterology) will achieve the best outcomes. Additionally, the incorporation of highly effective CFTR modulator therapies in early life may be one such way to alter both long-term outcomes and the relationship between early-life and adult height achievement.

Linear growth correlates closely with lung growth in people with CF; therefore, identifying both risk factors and effective preventative treatments for linear growth failure could lead to improved overall CF patient health. If new CFTR modulator therapies are beneficial to overall growth (weight and height), nutritional guidelines may need modification to address overall health outcomes. Encouraging calorie intake and high-fat diets may not continue to be advantageous, given the balance with obesity, fat mass versus lean body mass, and the potential for the development of CFRD.

## Figures and Tables

**Figure 1 nutrients-13-04414-f001:**
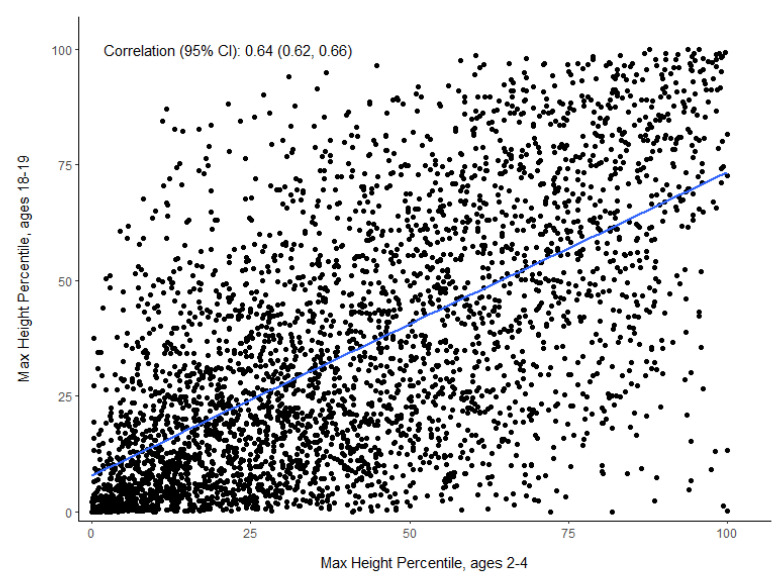
Maximum annualized HFA% in CFFPR between ages of 18 and 19 versus maximum annualized HFA% between ages 2 and 4 years (*n* = 3566).

**Table 1 nutrients-13-04414-t001:** Height percentile grouping at 2–4 years of age in CFFPR versus height percentile grouping at age 18–19 years of age. The individuals lowest annual HFA% in the 2- or 3-year windows, adult and childhood, respectively, defined their inclusion into categories.

Age 2–4 Height Percentile	Age 18–19 Height Percentile *n* % of Row				Total *n* % of Column
	<10th	10th–25th	25th–50th	≥50th	
<10th	563 54.4%	283 27.4%	150 14.5%	38 3.7%	1034 28.3%
10th–25th	231 23.8%	327 33.7%	285 29.4%	126 13%	969 26.5%
25th–50th	97 10.8%	207 22.9%	323 35.8%	275 30.5%	902 24.7%
≥50th	24 3.2%	63 8.4%	163 21.7%	500 66.7%	750 20.5%
Total	915 25%	880 24.1%	921 25.2%	939 25.7%	3655 100%

## Data Availability

Data were acquired though the CFF Patient Registry and are available to investigators upon request and scientific approval (https://www.cff.org/researchers/patient-registry-data-requests).

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
