# Peer review of "Most Short Children with Cystic Fibrosis Do Not Catch Up by Adulthood"

_nutrients, 2021, doi:10.3390/nu13124414_

Round 1
Reviewer 1 Report
Very well written paper looking at adult height. Agree that not having parental height is a shortcoming but the authors have dealt with this issue very nicely. I like the discussion looking at intrinsic factors that may explain this height deficit.
Author Response
Reviewer 1:
Very well written paper looking at adult height. Agree that not having parental height is a shortcoming but the authors have dealt with this issue very nicely. I like the discussion looking at intrinsic factors that may explain this height deficit.
We thank the reviewer for the compliments and feedback on the manuscript. We echo their enthusiasm on the topic.

Reviewer 2 Report
This manuscript is a short communication presenting the results of a retrospective case control study investigating how height in early childhood correlates with later height at age 18-19 using the Cystic Fibrosis Foundation Patient registry. This is a very important and understudied area in CF. This manuscript is well written and represents a valuable addition to the medical literature. Strengths include the large number of people studied, the prospective nature of data collection through the CFFPR, and the comprehensive discussion placing results in context of what is currently known on this topic. Concerns/suggestions are relatively minor, as below:
- The title “Short Children with CF Don’t Catch Up by Adulthood” seems a little leading, especially since almost half (45%) of those <10th percentile at 2-4y did in fact end up >10th percentile at 18-19y - one could argue that these kids technically did have some catch up by adulthood.
- ‘CFF’ needs to be defined in the Abstract on line 15.
- Much of the discussion focuses on the role of nutrition in growth, including how median BMI is now >50th percentile in children with CF but height lags; it would be helpful to include early childhood BMI results in this manuscript given that the data extend back into the 1990s when kids’ BMIs may not have been as good as they are now. It might also be interesting to include early childhood BMI Z-score as a covariate in the correlation between early childhood and adult height Z-scores.
- I am curious if the authors investigated those with very short stature (i.e. <3rd percentile) and if the height correlations in this group could be even stronger?
- Although this may be outside the scope of this manuscript, I can’t help but wonder about those patients who started out with a low childhood height percentile but then ended up with height >25th or >50th percentile – could we use these data to predict which patients may end up having the most catch-up growth? Could gender affect these results?
- Delayed puberty has been reported as more common in CF than the general population. It is possible that using the lowest height percentile between the ages of 18-19 for table 1 categories may be capturing some with delayed puberty that are still growing and have not yet reached their full adult height. Although probably not a huge number, this may lead to overestimation of the prevalence of adult short stature. Perhaps using the highest height percentile between 18-19y for table 1 could help address this?
- The authors appropriately point out important weaknesses of this study including the lack of comparison with healthy children and the absence of mid-parental height data. However, there are other limitations that should be acknowledged:
- The authors nicely lay out the potential impact of CFTR modulators on growth as well as ongoing studies investigating highly effective modulator therapy (HEMT) on growth and body composition. Presumably only a very small number of patients would have been treated with modulators in this cohort as only ivacaftor was available up until 2015. It is important to mention that given the advent of HEMT, analyzing results of childhood growth data from over 20 years ago may not be as applicable to today’s children in the post-modulator era.
- There are no data presented on other factors that may affect growth, such as oral or inhaled glucocorticoid use, growth hormone treatment, CFTR genotype and modulator use, pancreatic insufficiency, etc.
Author Response
Reviewer 2:
This manuscript is a short communication presenting the results of a retrospective case control study investigating how height in early childhood correlates with later height at age 18-19 using the Cystic Fibrosis Foundation Patient registry. This is a very important and understudied area in CF. This manuscript is well written and represents a valuable addition to the medical literature. Strengths include the large number of people studied, the prospective nature of data collection through the CFFPR, and the comprehensive discussion placing results in context of what is currently known on this topic. Concerns/suggestions are relatively minor, as below:
1. The title “Short Children with CF Don’t Catch Up by Adulthood” seems a little leading, especially since almost half (45%) of those <10th percentile at 2-4y did in fact end up >10thpercentile at 18-19y - one could argue that these kids technically did have some catch up by adulthood.
We agree with this observation; in response, we have now amended the title to read, “Most Short Children with Cystic Fibrosis Don’t Catch Up by Adulthood.”
2. ‘CFF’ needs to be defined in the Abstract on line 15.
We appreciate this suggestion and have done so.
3. Much of the discussion focuses on the role of nutrition in growth, including how median BMI is now >50th percentile in children with CF but height lags; it would be helpful to include early childhood BMI results in this manuscript given that the data extend back into the 1990s when kids’ BMIs may not have been as good as they are now. It might also be interesting to include early childhood BMI Z-score as a covariate in the correlation between early childhood and adult height Z-scores.
A big reason that BMI ‘looks so good’ in CF is because of short stature. CF care providers have made huge improvements on weight gain with nutritional interventions, but it does not impact height the way one might expect. The artifact of BMI being artificially high due to short stature is why we do not want to perpetuate it as if it was a valid marker in the CF population. To clarify this issue, we added explanatory text on lines 160-162 of the discussion.
4. I am curious if the authors investigated those with very short stature (i.e. <3rd percentile) and if the height correlations in this group could be even stronger?
We were reluctant to use an additional threshold (particularly one with less precedent) for risk of finding false associations due to multiple testing. For simplicity, we did not add this rationale to the current manuscript text.
5. Although this may be outside the scope of this manuscript, I can’t help but wonder about those patients who started out with a low childhood height percentile but then ended up with height >25th or >50th percentile – could we use these data to predict which patients may end up having the most catch-up growth? Could gender affect these results?
We agree that this is an interesting idea, and that it would be worth examining the features of those who ‘most caught up’ in a future, more extensive study; however, we also agree that this separate analysis is beyond the scope of this manuscript. Regarding the second question, we felt gender would be unlikely to impact the results because the percentiles used are adjusted for gender/sex; however, we agree that this question is also worth exploring. We have added text noting that these areas remain to be analyzed in future studies to the discussion (lines 264-66).
6. Delayed puberty has been reported as more common in CF than the general population. It is possible that using the lowest height percentile between the ages of 18-19 for table 1 categories may be capturing some with delayed puberty that are still growing and have not yet reached their full adult height. Although probably not a huge number, this may lead to overestimation of the prevalence of adult short stature. Perhaps using the highest height percentile between 18-19y for table 1 could help address this?
The author brings up another interesting point that we now briefly address in the discussion (lines 259-261) and may be worthy of additional analysis in a subsequent manuscript.
7. The authors appropriately point out important weaknesses of this study including the lack of comparison with healthy children and the absence of mid-parental height data. However, there are other limitations that should be acknowledged:
-
- The authors nicely lay out the potential impact of CFTR modulators on growth as well as ongoing studies investigating highly effective modulator therapy (HEMT) on growth and body composition. Presumably only a very small number of patients would have been treated with modulators in this cohort as only ivacaftor was available up until 2015. It is important to mention that given the advent of HEMT, analyzing results of childhood growth data from over 20 years ago may not be as applicable to today’s children in the post-modulator era.
- There are no data presented on other factors that may affect growth, such as oral or inhaled glucocorticoid use, growth hormone treatment, CFTR genotype and modulator use, pancreatic insufficiency, etc.
As above, we agree that these details are all worth noting and examining in a comprehensive future analysis. We briefly note this in the discussion (lines 261-264).

This manuscript is a resubmission of an earlier submission. The following is a list of the peer review reports and author responses from that submission.